# Yolov5 Series Algorithm for Road Marking Sign Identification

**Christine Dewi** [1],*[iD] **, Rung-Ching Chen** [2],*[iD]**, Yong-Cun Zhuang** [2] **and Henoch Juli Christanto** [3]

1    Department of Information Technology, Satya Wacana Christian University, Salatiga 50711, Indonesia
2    Department of Information Management, Chaoyang University of Technology, Taichung 41349, Taiwan
3    Department of Information System, Atma Jaya Catholic University of Indonesia, Jakarta 12930, Indonesia
*    Correspondence: christine.dewi@uksw.edu (C.D.); crching@cyut.edu.tw (R.-C.C.);

**Abstract:** Road markings and signs provide vehicles and pedestrians with essential information that assists them to follow the traffic regulations. Road surface markings include pedestrian crossings, directional arrows, zebra crossings, speed limit signs, other similar signs and text, and so on, which are usually painted directly onto the road surface. Road markings fulfill a variety of important functions, such as alerting drivers to the potentially hazardous road section, directing traffic, prohibiting certain actions, and slowing down. This research paper provides a summary of the Yolov5 algorithm series for road marking sign identification, which includes Yolov5s, Yolov5m, Yolov5n, Yolov5l, and Yolov5x. This study explores a wide range of contemporary object detectors, such as the ones that are used to determine the location of road marking signs. Performance metrics monitor important data, including the quantity of BFLOPS, the mean average precision (*mAP*), and the detection time (*IoU*). Our findings shows that Yolov5m is the most stable method compared to other methods with 76% precision, 86% recall, and 83% *mAP* during the training stage. Moreover, Yolov5m and Yolov5l achieve the highest score, *mAP* 87% on average in the testing stage. In addition, we have created a new dataset for road marking signs in Taiwan, called TRMSD.

**Keywords:** Yolov5; road marking; machine learning; object recognition

## 1. Introduction

Traffic signs are used to control traffic, alert drivers to potential danger ahead providing useful information to make driving safer and easier [1]. To provide unambiguous information, traffic signs are designed using several fundamentally distinct design styles. In addition, the background with many different buildings and shop signs makes it hard for the system to identify the street signs automatically. Thus, it becomes difficult to locate the road signs in many environments [2]. Road markings alert drivers to dangerous areas on the road, indicate traffic directions, suggest they slow down, and perform other useful tasks. There are various types of landmarks that are used for vehicle localization purposes, among which one of the most common types of landmarks is road surface markings.

Automatic detection of road markings has been used for autonomous vehicle guidance [3] or for lane detection systems [4], which is a common active safety feature of commercial cars. In recent years, various strategies for identifying road signs have been developed [5,6]. Most of the strategies deal with single images with live backgrounds [7]. Ref. [8] described a method for detecting road signs using a color index system. The first step in the process of designing road signs involved depicting each model in terms of a color histogram. An investigation on four distinct varieties of road markings was carried out by Qin and colleagues [9] using a machine vision approach. The images of the marking outlines were then retrieved from the photographs after being created at random using image processing techniques.

Due to the high computational capability, various deep learning methods were recently proposed for recognizing road markers. Following the calculation of the geometric

characteristics and pixel distribution of road markings by Soilan et al. [10], a two-layer feedforward network was used to extract pedestrian crossings and five different arrow classes. In the study by Wen et al. [11], a pixel-level U-net segmentation network was created to extract road markings. Chen et al. [12] introduced a deep learning model based on a dense feature pyramid network, which, to extract the path markers, was trained from start to finish. Cheng et al. [13] proposed using road markings as an intensity threshold technique based on normalization of unsupervised intensity and a deep learning strategy [14].

Current state-of-the-art detection frameworks, such as single shot detector (SSD) [15] and You Only Look Once (Yolo) [16], can be inferred in real time while maintaining robustness for the task of road marking detection. Yolo has demonstrated an impressively quick detection speed, even though sacrificing its accuracy. Since then, Joseph et al. made a few minor adjustments and came up with Yolov2, Yolov3, Yolov4 [17], and Yolov5 [18] as a result. Yolo V4, based on cross stage partial network (CSPNet) [19], was proposed for object detection. In that research, a network scaling approach was used to modify the depth, width, and resolution of the network, as well as the network's topology, resulting in the construction of Scaled-Yolo V4. Yolo V4 is designed for real-time object detection on general GPU. To obtain the best speed and accuracy trade-off, C.Y. Wang et al. [19] re-designed Yolo V4 to Yolo V4-CSP.

The Yolov5 model that was made available to the public in May of 2020 has several advantages. It is lightweight, quick, and able to run on the mobile platform. Additionally, it can support a greater variety of applications. As a result, the purpose of this experiment is to evaluate Yolov5, which is supposed to increase both the accuracy and the efficiency. The model's architecture is entirely determined by the model-configurations file. The *ultralytics* are compatible with a variety of Yolov5 designs and these models are referred to collectively as P5. Yolov5 models are distinguished from one another principally by the size of the parameters that they make use of, which are as follows: Yolov5n (nano), Yolov5s (small), Yolov5m (medium), Yolov5l (large), and Yolov5x (extra-large). When using this design for teaching purposes, it is advised to use images with a resolution of 640 pixels by 640 pixels.

The main contribution of this research is as follows: (1) The Yolov5 series of object identification algorithms, including Yolov5s, Yolov5m, Yolov5n, Yolov5l, and Yolov5x, are summarized in this research work. (2) This research investigates a variety of modern object detectors, including those used to identify road marking signs. Performance metrics track crucial data including the mean average precision (mAP), detection time (IoU), and the quantity of BFLOPS. (3) We create a new dataset called the Taiwan road marking sign dataset (TRMSD).

The organizational structure of the paper is as follows. In Section 2, we review the most recent publications in the field and describe our approach. Section 3 summarizes the experimental results. Section 4 focuses on discussing our findings, and Section 5 describes our conclusions and future research goals.

## 2. Materials and Methods

### 2.1. Road Marking Sign Identification

The identification of road markings is an essential task of intelligent transportation systems [20]. Road markings are as vital as road signs; not only do they help to provide a more secure environment for drivers but also provide drivers and other motorists with information that would not be able to be transmitted via traditional signs. Road markings can either be advisory, such as the reminder to keep the lines clear, or enforceable, such as yellow lines, box junctions, and stop lines. A large body of works has been created to tackle the challenge of automatically recognizing road markings. Previous researchers used various image processing approaches to identify road markings and signs. For instance, Foucher et al. [21,22] described a method for detecting and recognizing various painted road markings, including lanes, crosswalks, arrows, and other related road markings. These markings were all applied to the road. They presented a method for the recognition of road markings that was comprised of two stages: the first stage was the extraction of marking

elements, and the second stage was the identification of related components based on a single pattern or recurring rectangular patterns [23]. Another method for the detection and identification of road markings was presented by Ding et al. [24]. In order to recognize and categorize five different road markings, HOG characteristics and a support vector machine (SVM) were applied. The technique for recognizing symbol-based road markings that was published by Greenhalgh et al. [25] made use of HOG features in addition to a support vector machine.

*2.2. Yolov5 Architecture*

There are five distinct architectures for the Yolov5, including Yolov5s, Yolov5m, Yolov5n, and Yolov5l. The primary distinction is based on the quantity of feature extraction modules and convolution kernels that are dispersed over the network at various predetermined locations. Figure 1 presents a diagrammatic representation of the internal network that Yolov5 possesses. The Yolov5 design incorporates a number of different technologies, including automatic learning bounding box anchoring, mosaic data improvement, and cross-stage partial networking. This design makes use of the most effective algorithm optimization methods that were developed recently for convolutional neural networks. It is built on the Yolo detection architecture.

Yolov5 has four main components: input, backbone, neck, and output [26]. The backbone model's primary responsibility is to single out significant elements within the input image for analysis. When it comes to extracting rich, important characteristics from input photos, Yolov5 relies on cross stage partial networks (CSP) and spatial pyramid pooling (SPP) as its primary building blocks. SPP is beneficial for identifying the same item in multiple sizes and scales, which is important when it comes to the correct generalization of a model concerning object scaling. The feature pyramid architectures of the feature pyramid network (FPN) [27] and path aggregation network (PANet) [28,29] are utilized in the construction of the neck network. Powerful semantic features are distributed throughout the FPN structure, beginning at the top feature maps and working their way down to the lower feature maps. At the same time, it is the responsibility of the PAN structure to ensure the transmission of reliable localization features from lower feature maps to higher feature maps. PANet is utilized as a neck in Yolo v5, which allows for the generation of a feature pyramid [30,31].

Yolov5 is quite similar to Yolov4, although there are a few key differences between them: (1) Yolov4 is distributed using the Darknet framework, which is written in the programming language C. *PyTorch* serves as the underlying infrastructure for Yolov5. (2) In Yolov4, the configuration file is a *a.cfg* file, while in Yolov5, the configuration file is a *a.yaml* file. The cross stage partial network is the full name of CSP Net, which is a solution that primarily addresses the issue of a significant quantity of calculation in reasoning from the point of view of the design of network structures. According to the author of CSP Net, the issue of unnecessary inference calculations can be traced back to the network optimization gradient information repetition [32].

Using the structure of Yolov5 as an example, an initial image with dimensions of $608 \times 608 \times 3$ is input into the focus structure, as shown in Figure 1. Next, the slicing operation is used to change the image into a feature map with dimensions of $304 \times 304 \times 12$, and this is followed by 32 convolution operation kernels, which result in a final feature map with dimensions of $304 \times 304 \times 32$. The information presented in Table 1 provides an overview of all of Yolov5's models, including the inference speed on CPU and GPU, as well as the number of parameters with an image size of 640 [33,34].

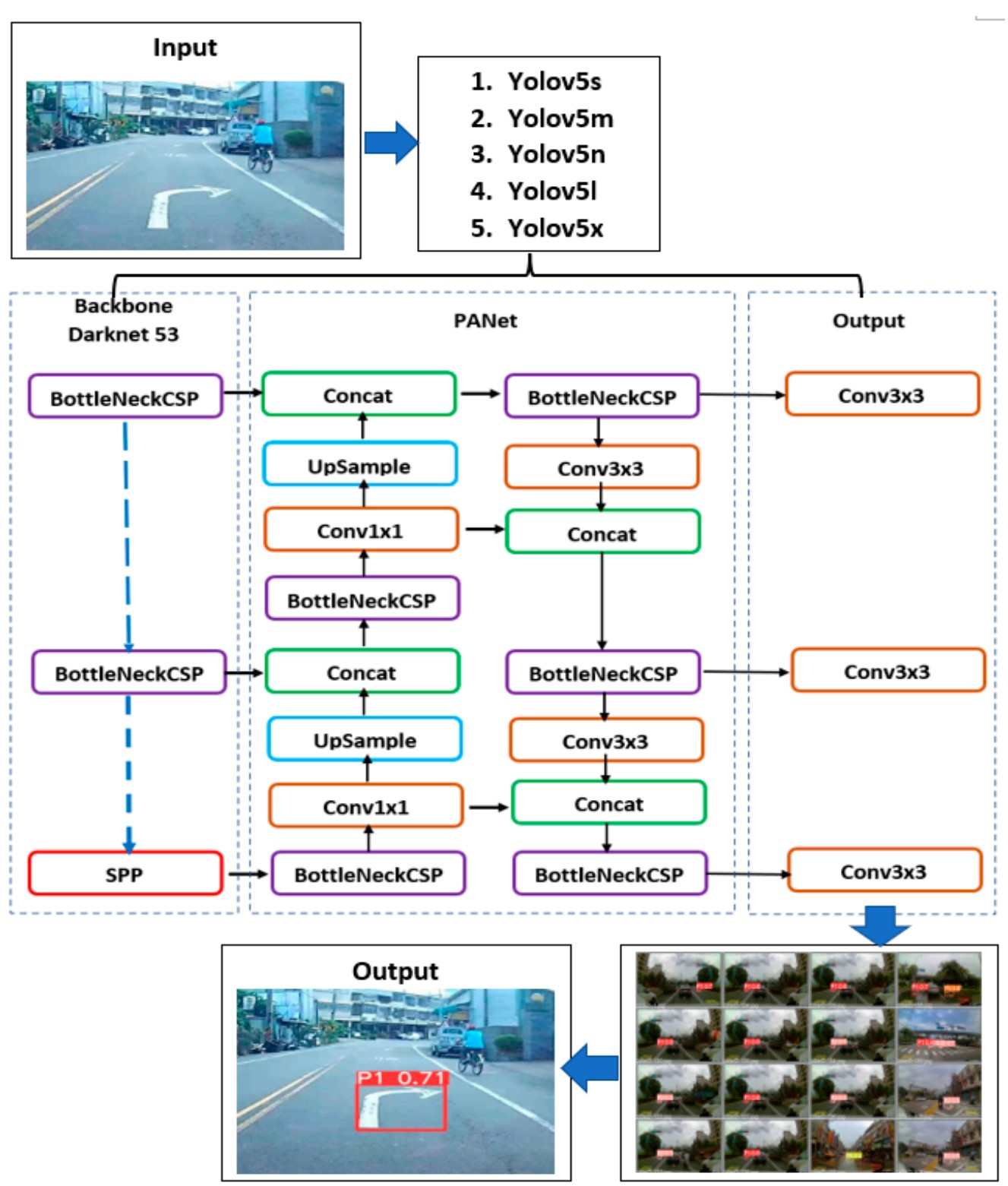

**Figure 1.** Yolov5 architecture.

**Table 1.** An overview of YoloV5 models.

| Model Name | Params (Million) | Accuracy (mAP 0.5) | CPU Time (ms) | GPU Time (ms) |
|---|---|---|---|---|
| Yolov5n | 1.9 | 45.7 | 45 | 6.3 |
| Yolov5s | 7.2 | 56.8 | 98 | 6.4 |
| Yolov5m | 21.2 | 64.1 | 224 | 8.2 |
| Yolov5l | 46.5 | 67.3 | 430 | 10.1 |
| Yolov5x | 86.7 | 68.9 | 766 | 12.1 |

The vast majority of annotation tools produce their results in the Yolo labeling format, which generates a single text file containing annotations for each image. Each text file has one bounding box, abbreviated as "BBox", with annotation for each of the objects that are displayed in the image. The annotations are scaled appropriately to the image, and their values range from 0 to 1 inclusively [35]. The following equations are the basis for the adjustment procedure for the Yolo format calculation.

$$dw = 1/W \tag{1}$$

$$x = \frac{(x_1 + x_2)}{2} \times dw \tag{2}$$

$$dh = 1/H \tag{3}$$

$$y = \frac{(y_1 + y_2)}{2} \times dh \tag{4}$$

$$w = (x_2 - x_1) \times dw \tag{5}$$

$$h = (y_2 - y_1) \times dh \tag{6}$$

$H$ indicates the height of the image, $dh$ refers to the absolute height of the image, $W$ serves as the width of the image, and $dw$ represents the absolute width of the picture.

Based on Table 1, Yolov5n is a newly announced nano model that is the smallest in the family. It is designed to be used for edge computing, Internet of Things devices, and has support for OpenCV deep neural networks. When saved in INT8 format, it is less than 2.5 MB, and when saved in FP32 format, it is roughly 4 MB. It works wonderfully for applications on mobile devices. Next, Yolov5s is the smallest model in the family and has approximately 7.2 million parameters. It is an excellent choice for performing inference on the CPU because of its tiny size. This is the medium-sized model, Yolov5m, which has a total of 21.2 million different parameters [36]. Due to the fact that it strikes a healthy mix between speed and accuracy, it is possibly the model that is most suited for the majority of datasets and training [37]. Moreover, the Yolov5l model has 46.5 million different parameters and is the largest member of the Yolov5 family. It works wonderfully for datasets in which we need to identify more discrete items. Furthermore, the Yolov5x is the largest of the five models, and it also has the highest *mAP* value of the five. It is considerably slower than the others and contains 86.7 million parameters [6].

## 3. Results

### 3.1. Taiwan Road Marking Sign Dataset (TRMSD)

In addition, we carried out this experiment utilizing images obtained from various video and image sources in order to simulate the road marking signs that are used in Taiwan. Eighty percent of images in the dataset are used for training, while the remaining twenty percent are used for validation. Figure 2 shows the Taiwan road marking sign dataset (TRMSD) sample image and Table 2 displays the TRMSD statistics. Furthermore, to prevent an imbalance in the data, we use images with a range of 391–409 for each class. Thus, our dataset consists of 6009 images in total, each with a size of 512 × 288.

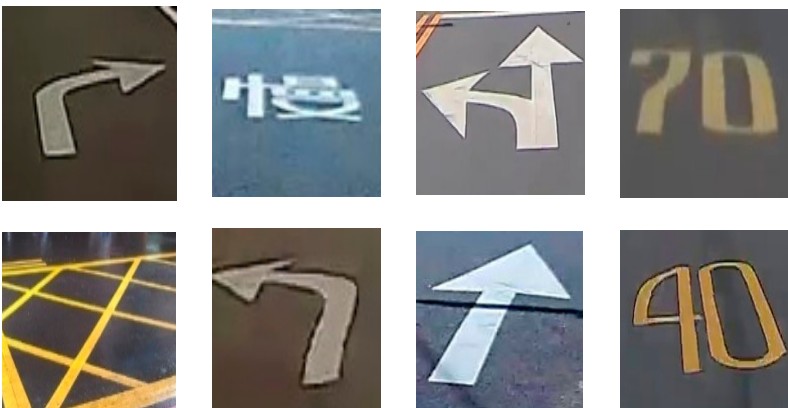

**Figure 2.** TRMSD sample image.

**Table 2.** TRMSD Dataset.

| Class ID | Class Name | Total Image |
|---|---|---|
| P1 | Turn Right | 405 |
| P2 | Turn Left | 401 |
| P3 | Go Straight | 407 |
| P4 | Turn Right or Go Straight | 409 |
| P5 | Turn Left or Go Straight | 403 |
| P6 | Speed Limit (40) | 391 |
| P7 | Speed Limit (50) | 401 |
| P8 | Speed Limit (60) | 400 |
| P9 | Speed Limit (70) | 398 |
| P10 | Zebra Crossing | 401 |
| P11 | Slow Sign | 399 |
| P12 | Overtaking Prohibited | 404 |
| P13 | Barrier Line | 409 |
| P14 | Cross Hatch | 398 |
| P15 | Stop Line | 403 |

Our experiment consisted of 15 classes (P1–P15), including Go Straight Turn Left, Turn Right, Turn Right or Go Straight, Turn Left or Go Straight, Speed Limit 40, Speed Limit 50, Speed Limit 60, Speed Limit 70, Zebra Crossing, Slow Sign, Overtaking Prohibited, Barrier Line, Cross Hatch, and Stop Line.

*3.2. Training Result*

The training process and result are explained in this section. We divided our dataset into a training subset and a testing subset: 70 percent for training and 30 percent for testing. Figure 3 describes the training process for batch 0 and batch 1. The anchor boxes are generated by Yolov5 with the help of a genetic algorithm. This procedure is referred to as the auto anchor process, which recalculates the anchor boxes in order to better match the data in the event that the default ones are inadequate. To generate k-Means evolved anchor boxes, this is combined with the k-Means algorithm. This is one of the reasons that Yolov5 performs so admirably even on a wide variety of datasets. Figure 4 describes the Yolov5 validation process for batch 0 and batch 1. During the training phase of Yolov5, four individual images are spliced to form a larger image. Each of the four individual images is subject to a random processing step during the splicing phase, which results in varying dimensions and configurations for each individual image. To make an analysis of our model, we will use the validation script. Using the "task" option, one can decide whether the performance is evaluated across the training dataset, the validation dataset, or the test dataset.

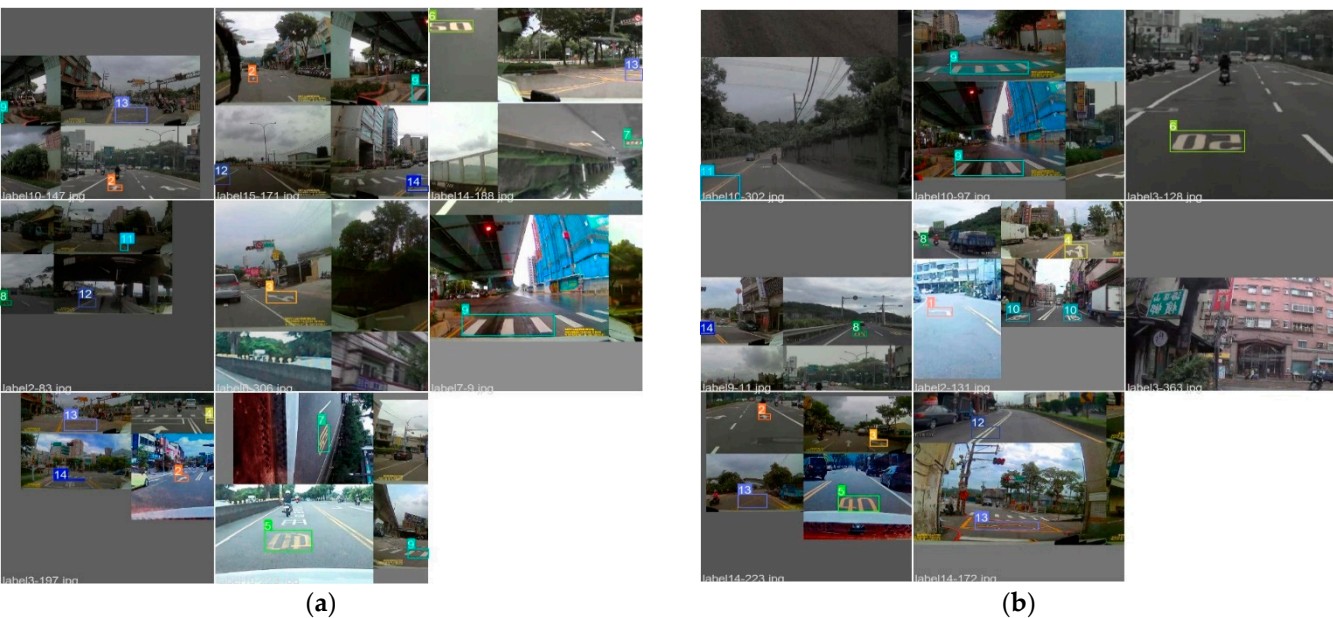

**Figure 3.** Training Process, (**a**) Batch 0 and (**b**) Batch 1.

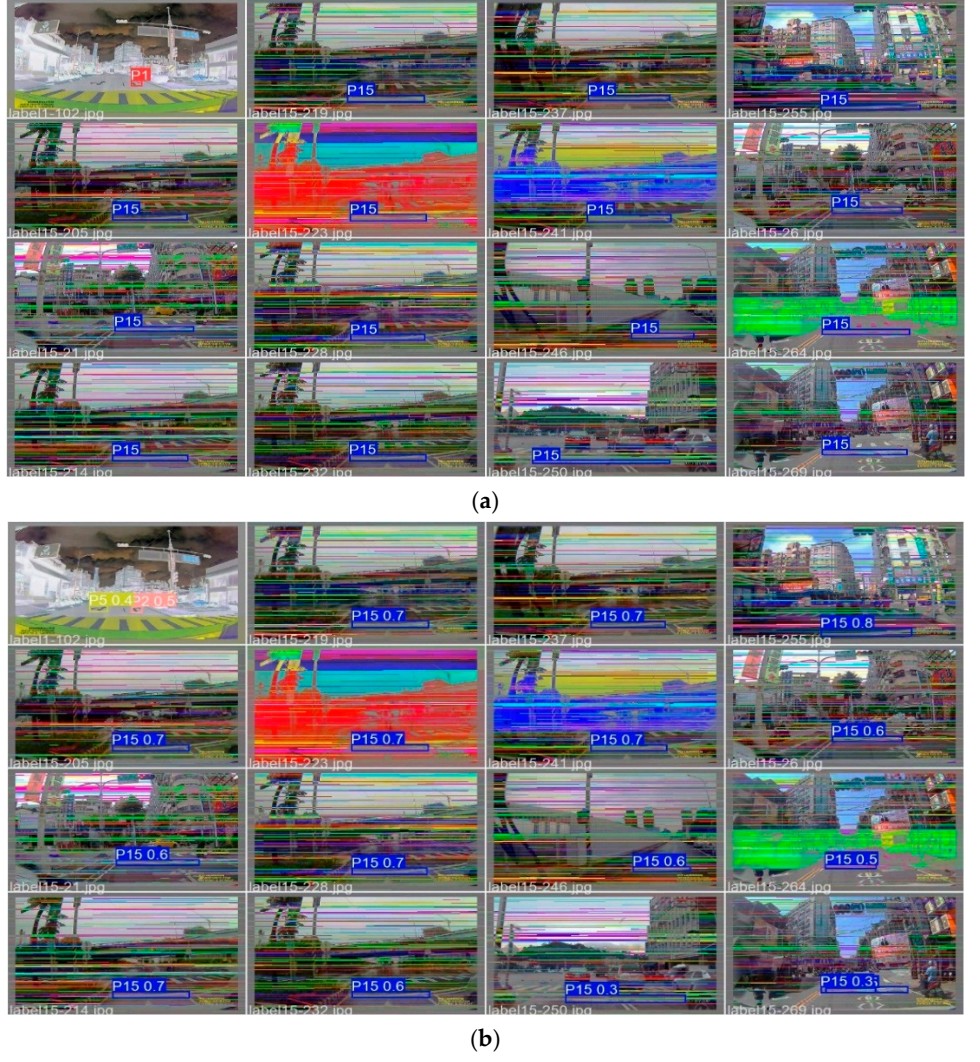

**Figure 4.** Validation Process batch 0. (**a**) Labels and (**b**) Predictions.

By default, all the results are logged into the runs/train directory, and a new experiment directory is created for each subsequent training, with the names runs/train/exp2, runs/train/exp, etc. We can view the mosaics, labels, predictions, and augmentation effects by looking at the train and val jpgs. It is important to note that an Ultralytics mosaic data loader is needed for training, and this device merges four photos into one mosaic while training. The weights are saved after 50 epochs during the training stage.

The last stage in training is called fine-tuning, which is completely optional. It involves unfreezing the entire model that we obtained in the previous step and then retraining it on our data using a very small learning rate. This has the potential to achieve considerable improvements by gradually adjusting the previously trained features on the fresh data. The hyperparameters-configurations file is where the learning rate can be modified to suit our needs. For the sake of demonstrating the tutorial, we will use the hyperparameters that are defined in the built-in hyp.finetune.yaml file. These hyperparameters have a significantly lower learning rate than the default. The weights will initially be set to the values that were saved on the stage before this one. Once finished training, it is stored using the standard PyTorch protocol, which entails using the.pt file extension.

Transfer learning is an efficient method for retraining a model on new data without retraining the entire network. Instead, a portion of the initial weight is assigned, while the remainder is used to calculate losses and is modified by the optimizer. It uses fewer resources than standard training and allows for shorter training periods, although it can result in a decrease in the accuracy of the units being trained. In our model, we use transfer learning while training with the Yolov5 series model.

During the training phase, we watch the mAP@0.5 to assess how well our detector performs on the validation set; a greater number indicates better performance. The dataset yet another markup language (YAML) file is possibly one of the most essential aspects of the Yolov5 training. This file provides the class names with the path to the data that is used for training and validation. For the training script to be able to detect the image paths, the label paths, and the class names, we need to supply this file path as an argument when we are running the training script. The information is already included in the dataset. Table 3 describes the training process of the Yolov5 series. Yolov5m, Yolov5l, and Yolov5x achieve the same average of *mAP* at approximately 83%, followed by Yolov5s with an average of *mAP* 81%. Based on the training results, it can be concluded that Yolov5m is the most stable method compared to other methods with 76% precision, 86% recall, and 83% *mAP*. Furthermore, Class P6 and Class P9 achieve the highest accuracy for all models with *mAP* in the range of 99% to 100%.

**Table 3.** Training Performance of Yolov5 series.

| Class | Yolov5n | | | Yolov5s | | | Yolov5m | | | Yolov5l | | | Yolov5x | | |
|---|---|---|---|---|---|---|---|---|---|---|---|---|---|---|---|
| | P | R | mAP@.5 | P | R | mAP@.5 | P | R | mAP@.5 | P | R | mAP@.5 | P | R | mAP@.5 |
| P1 | 0.63 | 0.89 | 0.79 | 0.66 | 0.87 | 0.77 | 0.63 | 0.87 | 0.80 | 0.61 | 0.87 | 0.80 | 0.61 | 0.87 | 0.78 |
| P2 | 0.62 | 0.77 | 0.74 | 0.66 | 0.69 | 0.74 | 0.65 | 0.78 | 0.72 | 0.64 | 0.74 | 0.73 | 0.91 | 0.79 | 0.70 |
| P3 | 0.53 | 0.75 | 0.64 | 0.55 | 0.75 | 0.62 | 0.60 | 0.83 | 0.72 | 0.54 | 0.82 | 0.72 | 0.60 | 0.78 | 0.74 |
| P4 | 0.45 | 0.63 | 0.61 | 0.41 | 0.65 | 0.60 | 0.43 | 0.77 | 0.65 | 0.40 | 0.62 | 0.61 | 0.40 | 0.73 | 0.62 |
| P5 | 0.42 | 0.62 | 0.55 | 0.37 | 0.52 | 0.46 | 0.42 | 0.71 | 0.57 | 0.41 | 0.84 | 0.55 | 0.37 | 0.72 | 0.48 |
| P6 | 0.99 | 1.00 | 1.00 | 0.99 | 1.00 | 1.00 | 0.99 | 1.00 | 1.00 | 0.99 | 1.00 | 1.00 | 0.99 | 1.00 | 1.00 |
| P7 | 0.81 | 0.89 | 0.91 | 0.82 | 0.87 | 0.89 | 0.89 | 0.88 | 0.90 | 0.78 | 0.86 | 0.90 | 0.79 | 0.89 | 0.90 |
| P8 | 0.84 | 0.98 | 0.97 | 0.82 | 0.98 | 0.97 | 0.87 | 1.00 | 0.97 | 0.90 | 0.99 | 0.98 | 0.89 | 1.00 | 0.98 |
| P9 | 0.99 | 1.00 | 1.00 | 0.99 | 1.00 | 1.00 | 0.99 | 1.00 | 1.00 | 0.98 | 1.00 | 1.00 | 0.99 | 1.00 | 1.00 |
| P10 | 0.95 | 0.64 | 0.89 | 0.83 | 0.59 | 0.75 | 0.93 | 0.80 | 0.89 | 0.95 | 0.76 | 0.91 | 0.92 | 0.83 | 0.91 |
| P11 | 0.84 | 0.99 | 0.99 | 0.89 | 0.95 | 0.98 | 0.86 | 0.99 | 0.99 | 0.88 | 0.99 | 0.99 | 0.92 | 0.99 | 0.99 |
| P12 | 0.78 | 0.74 | 0.73 | 0.66 | 0.78 | 0.74 | 0.69 | 0.75 | 0.71 | 0.69 | 0.74 | 0.70 | 0.75 | 0.78 | 0.76 |
| P13 | 0.64 | 0.75 | 0.73 | 0.58 | 0.68 | 0.63 | 0.76 | 0.81 | 0.78 | 0.72 | 0.82 | 0.79 | 0.70 | 0.84 | 0.78 |
| P14 | 0.81 | 0.85 | 0.86 | 0.792 | 0.84 | 0.84 | 0.87 | 0.91 | 0.89 | 0.88 | 0.88 | 0.88 | 0.86 | 0.85 | 0.87 |
| P15 | 0.79 | 0.66 | 0.79 | 0.86 | 0.71 | 0.81 | 0.84 | 0.82 | 0.90 | 0.91 | 0.86 | 0.93 | 0.83 | 0.84 | 0.91 |
| All | 0.74 | 0.81 | 0.81 | 0.72 | 0.79 | 0.79 | 0.76 | 0.86 | 0.83 | 0.75 | 0.82 | 0.83 | 0.75 | 0.86 | 0.83 |

Additionally, we can acquire the precision–recall curve, which is automatically saved for every validation. Figure 5 depicts the precision and recall for Yolov5m and Yolov5l. The performance metrics we use to evaluate the performance of our dataset TRMSD in Yolov5 model experiments include precision, recall, accuracy score, and F1. Among them, precision and recall are defined in Equations (7) and (8) [38], then accuracy and F1 are defined in Equations (9) and (10) [39]. A compound loss is computed for the Yolo family of algorithms, with the score for objectiveness, the score for class probability, and the score for bounding box regression serving as the inputs. For the purpose of calculating the loss of class probability and object score, Ultralytics uses the binary cross-entropy with logits loss function that is available in PyTorch [40]. True positive (TP) is the number of "yes" in the real situation when the model evaluation is also "yes", and true negative (TN) is the number of "no" in the real situation when the model evaluation is also "no". False positive (FP) is the number of "no" in the real situation when the model evaluation is also "yes"; false negative (FN) is the number of "yes" in the real situation when the model evaluation is also "no" [41]. The mean average precision (mAP) is a popular indicator for evaluating the performance of object identification models and defined in Equation (11).

$$Precision\ (P) = \frac{TP}{TP + FP} \tag{7}$$

$$Recall\ (R) = \frac{TP}{TP + FN} \tag{8}$$

$$Accuracy\ (Acc) = \frac{TP + TN}{TP + FN + FP + FN} \tag{9}$$

$$F1 = \frac{2 \times Precision \times Recall}{Precision + Recall} \tag{10}$$

$$mAP = \frac{1}{N} \sum_{i=1}^{N} Acc \tag{11}$$

The Yolo loss function is calculated using the Equation (12) [16].

$$
\begin{aligned}
Yolo\ Loss\ Function = {} & \lambda_{coord} \sum_{i=0}^{s^2} \sum_{j=0}^{B} \mathbb{1}_{ij}^{obj} \left[ (x_i - \hat{x}_i)^2 + (y - \hat{y}_i)^2 \right] \\
& + \lambda_{coord} \sum_{i=0}^{s^2} \sum_{j=0}^{B} \mathbb{1}_{ij}^{obj} \left[ \left(\sqrt{w_i} - \sqrt{\hat{w}_i}\right)^2 + \left(\sqrt{h_i} - \sqrt{\hat{h}_i}\right)^2 \right] \\
& + \sum_{i=0}^{s^2} \sum_{j=0}^{B} \mathbb{1}_{ij}^{obj} (C_i - \hat{C}_i)^2 + \lambda_{noobj} \sum_{i=0}^{s^2} \sum_{j=0}^{B} \mathbb{1}_{ij}^{noobj} (C_i - \hat{C}_i)^2 \\
& + \sum_{i=0}^{s^2} \mathbb{1}_{i}^{obj} \sum_{c \in classes} (p_i(c) - \hat{p}_i(c))^2
\end{aligned}
\tag{12}
$$

where $S$ is the number of cells in the image, $B$ is the number of bounding boxes predicted in each grid cell, and c represents the class prediction for each grid cell. Furthermore, $p_i(c)$ represents the confidence probability. For any box $j$ of cell $i$, $x_{ij}$ and $y_{ij}$ represent the coordinates of the center of the anchor box, $h_{ij}$ gives height, $w_{ij}$ gives width of the box and $C_{ij}$ gives the confidence score. Finally, $\lambda coord$ and $\lambda noobj$ are the weights to decide the importance of localization and recognition in the training.

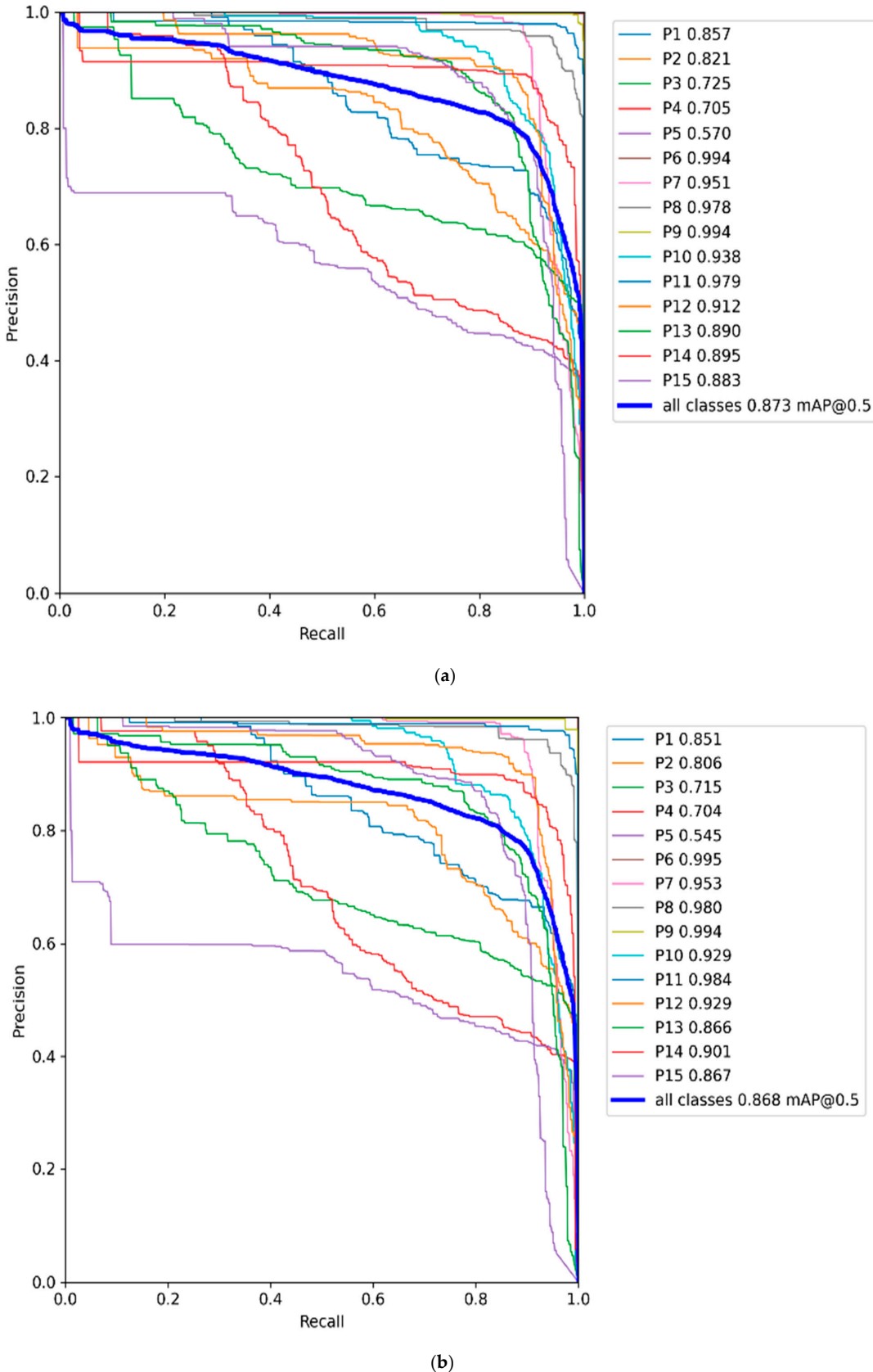

**Figure 5.** Precision and recall. (**a**) Yolov5m, and (**b**) Yolov5l.

## 4. Discussion

Table 4 describes the testing performance of the Yolov5 series. After achieving satisfactory results during the training phase, our model is now prepared for the inference phase. Each image is augmented (with a horizontal flip and three different resolutions), and the final prediction is an ensemble of all these augmentations. After inference, we can further improve the accuracy of the predictions by adding test-time augmentations (TTA). Because the TTA results in an inference that is two to three times longer, we have to abandon it to maintain a high frames-per-second (FPS) rate. An inference can be made based on an image, a video, a directory, a stream, a camera, or even a link to YouTube. These can all serve as the input. In our experiment, we test the Yolov5 series with some groups of images for each class. Based on our experiment results, Yolov5m and Yolov5l achieve the highest average mAP 87%, followed by Yolov5s with the mAP 84%, Yolov5x with mAP 83%, and Yolov5 with mAP 80%. Class P6 achieves the highest mAP for all Yolov5 series ranging from 99% to 100%.

**Table 4.** Testing Performance of Yolov5 series.

| Class | Yolov5n | | | Yolov5s | | | Yolov5m | | | Yolov5l | | | Yolov5x | | |
|---|---|---|---|---|---|---|---|---|---|---|---|---|---|---|---|
| | P | R | mAP@.5 | P | R | mAP@.5 | P | R | mAP@.5 | P | R | mAP@.5 | P | R | mAP@.5 |
| P1 | 0.62 | 0.92 | 0.81 | 0.64 | 0.92 | 0.84 | 0.65 | 0.93 | 0.86 | 0.65 | 0.94 | 0.85 | 0.61 | 0.87 | 0.78 |
| P2 | 0.69 | 0.67 | 0.75 | 0.72 | 0.75 | 0.79 | 0.77 | 0.73 | 0.82 | 0.79 | 0.73 | 0.81 | 0.91 | 0.79 | 0.70 |
| P3 | 0.54 | 0.74 | 0.62 | 0.55 | 0.80 | 0.67 | 0.60 | 0.89 | 0.73 | 0.60 | 0.79 | 0.72 | 0.60 | 0.78 | 0.74 |
| P4 | 0.51 | 0.59 | 0.64 | 0.50 | 0.69 | 0.67 | 0.51 | 0.75 | 0.71 | 0.54 | 0.66 | 0.70 | 0.40 | 0.73 | 0.62 |
| P5 | 0.48 | 0.65 | 0.53 | 0.49 | 0.69 | 0.55 | 0.51 | 0.65 | 0.57 | 0.51 | 0.64 | 0.55 | 0.37 | 0.72 | 0.48 |
| P6 | 0.99 | 1.00 | 1.00 | 0.99 | 1.00 | 1.00 | 1.00 | 1.00 | 0.99 | 0.99 | 1.00 | 1.00 | 0.99 | 1.00 | 1.00 |
| P7 | 0.84 | 0.91 | 0.93 | 0.87 | 0.91 | 0.96 | 0.93 | 0.90 | 0.95 | 0.88 | 0.91 | 0.95 | 0.79 | 0.89 | 0.90 |
| P8 | 0.82 | 0.97 | 0.96 | 0.83 | 0.97 | 0.97 | 0.86 | 0.99 | 0.98 | 0.82 | 0.99 | 0.98 | 0.89 | 1.00 | 0.98 |
| P9 | 0.98 | 1.00 | 0.99 | 0.98 | 1.00 | 0.99 | 0.98 | 1.00 | 0.99 | 0.98 | 1.00 | 0.99 | 0.99 | 1.00 | 1.00 |
| P10 | 0.90 | 0.61 | 0.82 | 0.87 | 0.66 | 0.87 | 0.93 | 0.80 | 0.94 | 0.88 | 0.80 | 0.93 | 0.92 | 0.83 | 0.91 |
| P11 | 0.80 | 0.99 | 0.97 | 0.89 | 0.99 | 0.98 | 0.91 | 1.00 | 0.98 | 0.89 | 1.00 | 0.98 | 0.92 | 0.99 | 0.99 |
| P12 | 0.78 | 0.69 | 0.76 | 0.71 | 0.86 | 0.81 | 0.86 | 0.89 | 0.91 | 0.88 | 0.92 | 0.93 | 0.75 | 0.78 | 0.76 |
| P13 | 0.71 | 0.66 | 0.70 | 0.68 | 0.83 | 0.80 | 0.77 | 0.88 | 0.89 | 0.80 | 0.85 | 0.87 | 0.70 | 0.84 | 0.78 |
| P14 | 0.73 | 0.83 | 0.82 | 0.80 | 0.86 | 0.85 | 0.86 | 0.92 | 0.90 | 0.88 | 0.88 | 0.90 | 0.86 | 0.85 | 0.87 |
| P15 | 0.84 | 0.56 | 0.72 | 0.88 | 0.67 | 0.80 | 0.92 | 0.70 | 0.88 | 0.90 | 0.71 | 0.87 | 0.83 | 0.84 | 0.91 |
| All | 0.75 | 0.79 | 0.80 | 0.76 | 0.84 | 0.84 | 0.80 | 0.87 | 0.87 | 0.8 | 0.85 | 0.87 | 0.75 | 0.86 | 0.83 |

When conducting deep learning, parameters known as hyperparameters are set in advance of formal training. The use of appropriate hyperparameters has the potential to increase the performance of the model. The Yolov5 method has a total of 23 hyperparameters, most of which are employed in the process of configuring the learning rate, loss function, and data improvement parameters and so on. It is discovered unequivocally that the more complex the network structure model, the lower the training loss convergence and the higher the validation loss, which indicates that the overfitting of the model is more severe. The complexity of the model can lead to an increase in the validation loss in proportional measure. The model's capacity to identify abnormalities shows just a slight improvement overall. The weight volume and the number of parameters are two measures that can be used to characterize the complexity of the model. The more complicated the model is, the higher these indexes are, and the more RAM the GPU requires to store it during training.

Figure 6 shows the recognition results of the Yolov5 series. The recognition results for Yolov5m are shown in Figure 6c. Class P1 achieved 73% accuracy, Class P14 obtained 68%, Class P3 showed accuracy of 66%, 83%, and 38%. All models can detect road markings sign in the image very well. The Yolov5 is a lightweight and relatively simple device to operate. Moreover, there are a total of 25 blocks in the Yolov5m medium model (from 0 to 24). Each block is composed of a stacked arrangement of a variety of layers. Yolov5 is quick to train, quick to draw conclusions, and does well in performance.

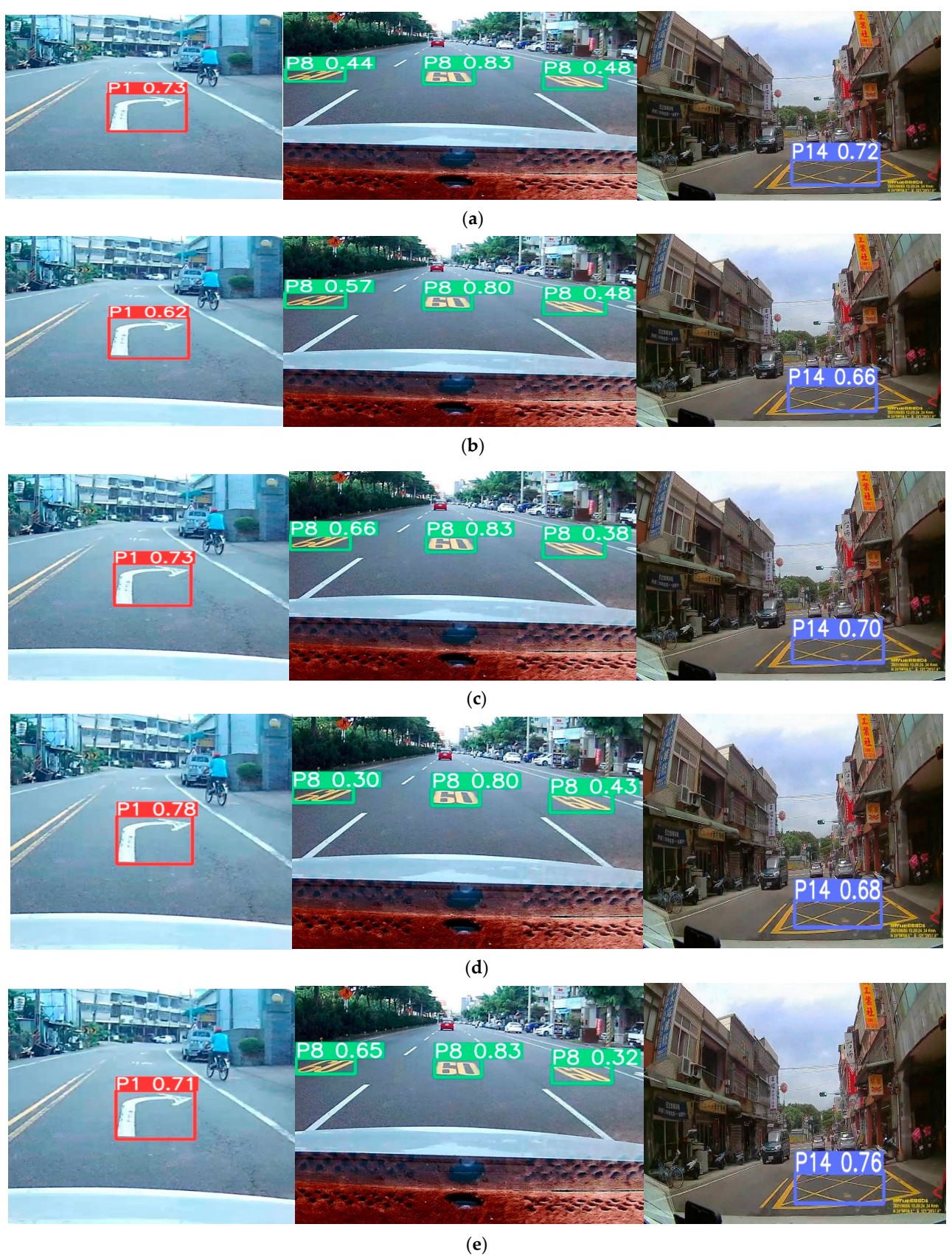

**Figure 6.** Recognition results. (**a**) Yolov5n, (**b**) Yolov5s, (**c**) Yolov5m, (**d**) Yolov5l, and (**e**) Yolov5x.

Figure 7 contains specific information regarding the detection performance and the level of complexity of a variety of models including layers, parameters, GFLOPS, and FPS. In our experiment, Yolov5x contains the most layers at approximately 444 layers, then Yolov5l has 367 layers, and Yolov5m, Yolov5s, Yolov5n have the same 213 layers. Yolov5x loads the most parameters at 86,267,620, and Yolov5m and Yolov5n contain the same parameters, 1,779,460. In our experiment, Yolov5s loads 24.6 FPS; Yolov5n, 40.5 FPS; Yolov5m, 11.4 FPS; Yolov5l, 2.8 FPS; and Yolov5x, 1.7 FPS.

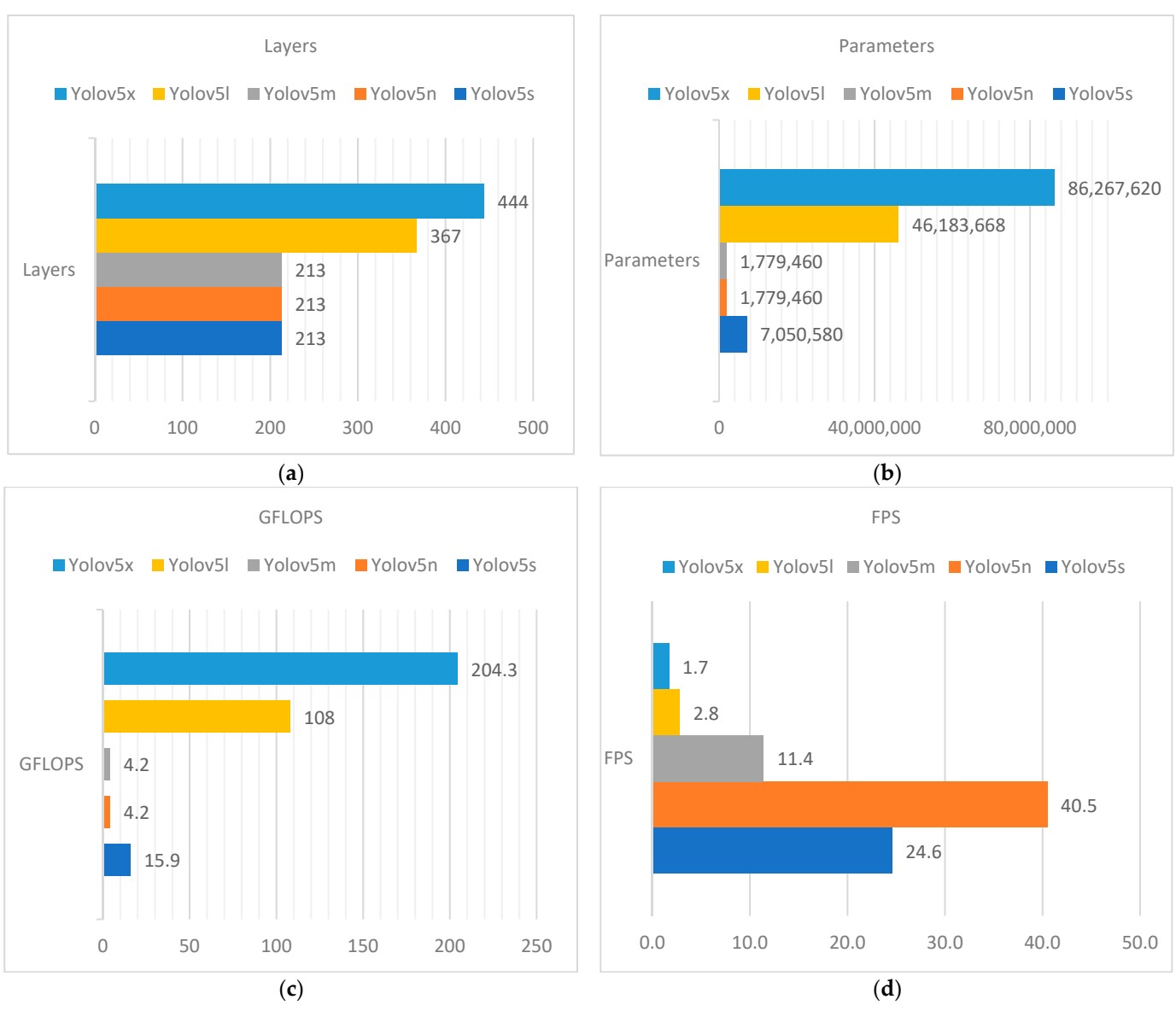

**Figure 7.** Comparison of the (**a**) Layers, (**b**) parameters, (**c**) GFLOPS, and (**d**) FPS.

The following is a list of benefits in using Yolov5: First, Yolo is the first model in the Yolo family that was originally written on PyTorch rather than on PJ Reddie's Darknet. Darknet is a research framework that offers a great degree of adaptability but was not developed with production situations in mind. It caters to a more limited population of end consumers. When all these factors are considered together, the result is that Darknets are more difficult to configure and less production ready. Because Yolov5 was originally implemented in PyTorch, Yolov5 reaps the benefits of the well-established ecosystem that PyTorch has to offer, including simplified support and simplified deployment. Moreover, as Yolov5 becomes more popular, the research community may find that iterations on Yolov5

become simpler. Because of this, deployment for mobile devices is also becoming easier, as the models can be easily compiled to ONNX or CoreML. Second, Yolov5 is incredibly fast, even by the current standards. Yolov5 is now capable of producing 140 frames per second in batches, and the Yolov5 implementation testing uses this setting by default. Yolov4 is capable of performing 30 FPS while Yolov5 only produces 10 FPS when the batch size is set to 1. Third, the Yolov5 formula is reliable. Although EfficientDet and Yolov4 yield similar results to our expectations, it is very unusual to observe such a broad increase in performance that is not accompanied by a reduction in accuracy. Fourth, Yolov5 has a low memory capacity. To be more specific, the weight file size used by Yolov5 is 27 megabytes. The weight file for Yolov4 (with Darknet architecture) takes up 244 MB of our storage space. When compared to Yolov4, Yolov5 is approximately 90 percent smaller. Because of this, Yolov5 can be applied to embedded devices in a much simpler way.

## 5. Conclusions

The objective of this study is to provide a concise overview of CNN-based object identification methods, with a particular focus on the Yolov5 algorithm series. During our experimental research, we tested and analyzed a variety of modern object detectors. Among the detectors, we investigated those that were designed to identify road marking signs. Important characteristics such as the mean average precisions (mAP), the detection time (IoU), and the number of BFLOPS were measured by the assessment criteria.

We have reached the following conclusion. First, Yolov5m is the most stable method compared to other methods with 76% precision, 86% recall, and 83% mAP during the training stage. Second, Yolov5m and Yolov5l achieve the highest average mAP 87% in the testing stage. Next, we created our own dataset for road marking signs in Taiwan (TRMSD). In future research, we plan to integrate the detection of road markings with explainable artificial intelligence (XAI). In addition, we are planning to upgrade our Taiwan road marking sign dataset (TRMSD) with an emphasis on the recognition of pothole signs, which can add different illuminations and textures conditions into our dataset.

**Author Contributions:** Conceptualization, C.D.; data curation, R.-C.C. and Y.-C.Z.; formal analysis, C.D.; investigation, C.D. and H.J.C.; methodology, C.D.; project administration, R.-C.C.; resources, Y.-C.Z.; software, Y.-C.Z. and H.J.C.; supervision, R.-C.C.; validation, C.D. and H.J.C.; visualization, H.J.C.; writing—original draft, C.D.; writing—review and editing, C.D. and R.-C.C. All authors have read and agreed to the published version of the manuscript.

**Funding:** This paper is supported by the Ministry of Science and Technology, Taiwan. The Nos are MOST-110-2927-I-324-50, MOST-110-2221-E-324-010, MOST-109-2622-E-324-004, and MOST-111-2622-E-324-002-Taiwan.

**Data Availability Statement:** The data presented in this study are available on request from the corresponding author.

**Acknowledgments:** The author would like to thank all colleagues from Chaoyang Technology University, Satya Wacana Christian University, Indonesia, Atma Jaya Catholic University of Indonesia, Jakarta, and all those involved in this research.

**Conflicts of Interest:** The authors declare no conflict of interest.

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
