# Peer review of "Yolov5 Series Algorithm for Road Marking Sign Identification"

_2504-2289, doi:10.3390/bdcc6040149_

Round 1
Reviewer 1 Report (Previous Reviewer 3)
Despite its overviewal nature, it is still valuable in assesment of Yolo framework application onto road signs/markings re-cognition. Inspite of its niche use on Taiwanesse markings set, it shows potential of application.
However, there are aspets that can be improved such as:
- Figure 7, a and b are completely useless for explanation! They are space fillers and doesn't explain anything usefull and what it is not said before. Don't practicise such things!
- Furthermore, there is no mention of any kind regarding training times. The experimenting platform is laid out but nothing regarding suitabiliy for real-time execution?
- Consequently to the previous comment, it will be more informative and valuable to express performantory aspets of Yolos on your road marking sign set using FPS or, at least, executory times for each of NN topology. Side-by-side to the fig. 7c will suit well!
- Furthermore, it is oftenly mixed/missused terms regarding detection precisions (xAP) and execution times (of any kind) that triggers suspect on article thorough readout and understanding of used methods.
Author Response
Please see the attachment.

Reviewer 2 Report (New Reviewer)
The contribution is too small for a journal paper.
The paper provides an evaluation of a given approach on a new (rather small) dataset. which is limited to one country and optimal lighting conditions. Thus, the useability of the dataset is limited. There is not much to be learnt from the paper, there is no new approach
Results are not shown in an appropriate way and are not surprising/interesting
Overall, it looks rather like a technical report than a journal contribution.
Author Response
Please see the attachment.

Reviewer 3 Report (New Reviewer)
The authors propose a “Yolov5 series algorithm for Road Marking Sign Identification”, which attempts to compare some computer vision algorithms applied to Road Marking Sign Identification. It is an interesting topic in the field and with a perspective of great evolution in the following years.
The authors correctly introduce the problem and give a good literature review of today’s algorithms applied to the topic.
After reading the paper I would like the authors comments related to:
-The authors in the abstract mention that will compare some algorithms in the Road Marking Sign Identification field, but only compare Yolo v5 versions…
-The dataset used to train, test and validation needs more images in each class. It was implemented any data augmentation strategy?
-The network was fully trained or was made any transfer learning and fine tuning? It’s not perceptible …
-Why are the low values in the classes P4 and P5 in all Yolo versions, as dataset is balanced?
-The present research work is interesting in the computer vision field, but from the paper itself I can’t conclude what the authors conclude, i.e., the experimental section must have a more complete approach.
As finally, can your evaluation be generalized to different illuminations and textures conditions with no lack of performance?
As a conclusion of the review, the authors are invited to address my questions. The paper is well written and for this reason my recommendation is that it may be accepted after corrections.
Round 2
Reviewer 2 Report (New Reviewer)
I don't see significant improvements in v2.
Author Response
Please see the attachment.

This manuscript is a resubmission of an earlier submission. The following is a list of the peer review reports and author responses from that submission.
Round 1
Reviewer 1 Report
The paper is well written, but it doesn't provide any contribution. Authors have only used existing and well-known models YOLO,and applied it to traditional problem traffic road signs.
Reviewer 2 Report
The manuscript entitled: " Road Marking Sign Recognition using Yolov5 series algorithm based on Deep Learning " presents very interesting scientific contents about detection road marking signs. In this study authors compared 5 versions of very popular object detection algorithm called YOLOv5 and they introduced a novel dataset of road marking signs. However, the presented manuscript contains several weak points, which need to be improved to be published in the journal:
· * In Table 1 and in the manuscript’s text, it is not written on which dataset, the mAP metric was evaluated,
· * The description of YOLOv5 may be not clear for a potential reader, who is not familiar with YOLO, I would recommend to start the description of this architecture comparing it to the YOLOv1. It would be important as YOLOv5 was not described in the technical paper by their authors.
· * The differences between various version of YOLOv5 are not well explained. I would recommend to extend the description in this place.
· * This sentence “Ultralytics is compatible with various Yolov5 designs” may be not clear for a potential reader”. What doea it mean “Ultralytics” in this context?
· * Quality of Figure 1 should be improved, e.g. the position how the arrows stick the boxes is not regular,
· * Term BottleNeckCSP fro Figure 1 is not explained,
· * The basic operations for YOLOv5 are not well explained,
· * Is the Taiwan Road Marking Sign Dataset (TRMSD) publicly available or it is planned their publication?
· * It is not clear how the road marking sign images were collected. What was the sensor setup to collect this dataset?
· * Why the images are used in so small resolution?
· * How the road marking sign were annotated? Did you have a specification defined for that?
· * The technical description of the implementation presented in lines 215-231 has limited value, and I would recommend to remove it from the manuscript.
· * What does mean the colors on images in Figure 4?
· * The presented results are not compared to the state-of-the-art object detectors.
· * In line 354 “mean acquisition time (mAP)” rather means mean average precision.
· * Computational time comparison is not presented.
· * The hardware setup on which computations were performed are not described,
Reviewer 3 Report
The article deals with everyinteresting topic of automated traffic sigh detection, recognition and classification. Here, authors utilize lately common interestly choice of NN variants that have oportunity in nonlinear-solving applications of imaginery processing and other multidimensional sets of data.
Despite of interesting topic and overally god presentation there are some aspects of the article content that can rise quality in general, so, as follows:
- Minor grammatical and typewriting errors are still present. Thorough proofreading will help greatly to improve readability and punctuality of the text.
- Explain the resons for choosing such particullary resolution of images and how does NN trainings behave using lower and higher resolution. Especially emphasize training times vs. mAP rate.
- Specify technical details of testing (and proving) platform that supports stated training and testing results in performative and qualitative manner.
- Specify training and finetuning times because GFLOPs doesn't mean anything without known training and testing enviroment and corresponding conditions.
- Update figure 7 with aditional bargraph that depicts FPS rate for each NN (in accordance with, previously clearly defined, testing and training environment and conditions)
Round 2
Reviewer 1 Report
The paper still lack of contribution
Reviewer 2 Report
The authors have provided detailed answers to my all comments and the manuscript can be accepted in the current form.